# The Development of a Sleep Intervention for Firefighters: The FIT-IN (Firefighter’s Therapy for Insomnia and Nightmares) Study

**DOI:** 10.3390/ijerph17238738

**Published:** 2020-11-24

**Authors:** Eun Hee Jang, Yujin Hong, Yeji Kim, Sangha Lee, Yeonsoon Ahn, Kyoung Sook Jeong, Tae-Won Jang, Hyejin Lim, Eunha Jung, Seockhoon Chung, Sooyeon Suh

**Affiliations:** 1Department of Psychology, Sungshin University, Seoul 02844, Korea; eunhee2914@gmail.com (E.H.J.); yuuugenie@gmail.com (Y.H.); yez109withyou@gmail.com (Y.K.); xrpsychology@gmail.com (S.L.); 2Department of Preventive Medicine, Yonsei University Wonju College of Medicine, Wonju 26426, Korea; ysahn1203@gmail.com; 3Department of Occupational and Environmental Medicine, Wonju Severance Christian Hospital, Wonju 26426, Korea; bandyoem@naver.com; 4Department of Occupational and Environmental Medicine, Hanyang University College of Medicine, Seoul 04763, Korea; om1024@hanmail.net; 5Ilsang Psychological Services, IPS, Seoul 05685, Korea; psych.imagine@gmail.com (H.L.); lotusrot@gmail.com (E.J.); 6Department of Psychiatry, Asan Medical Center, University of Ulsan College of Medicine, Seoul 05505, Korea

**Keywords:** firefighters, insomnia, nightmare, brief behavioral therapy for insomnia, imagery rehearsal therapy

## Abstract

*Background*: Firefighters are vulnerable to irregular sleep patterns and sleep disturbance due to work characteristics such as shift work and frequent dispatch. However, there are few studies investigating intervention targeting sleep for firefighters. This preliminary study aimed to develop and test a sleep intervention, namely FIT-IN (Firefighter’s Therapy for Insomnia and Nightmares), which was based on existing evidence-based treatment tailored to firefighters in consideration of their occupational characteristics. *Methods*: This study implemented a single-group pre-post study design, utilizing an intervention developed based on brief behavior therapy for insomnia with imagery rehearsal therapy components. FIT-IN consisted of a total of three sessions (two face-to-face group sessions and one telephone session). Participants were recruited from Korean fire stations, and a total of 39 firefighters participated. Participants completed a sleep diary for two weeks, as well as the following questionnaires to assess their sleep and psychological factors: insomnia severity index (ISI), disturbing dream and nightmare severity index (DDNSI), Epworth sleepiness scale (ESS), depressive symptom inventory-suicidality subscale (DSI), and Patient Health Questionnaire-9 (PHQ-9). These questionnaires were administered before the first session and at the end of the second session. *Results*: The FIT-IN program produced improvements in sleep indices. There was a significant increase in sleep efficiency (*p* < 0.01), and a decrease in sleep onset latency, number of awakenings, and time in bed (*p* < 0.05), as derived from weekly sleep diaries. In addition, significant decreases were shown for insomnia (*p* < 0.001) and nightmare severity (*p* < 0.01). *Conclusion*: There were significant improvements in sleep and other clinical indices (depression, PTSD scores) when comparing pre-and post-intervention scores. FIT-IN may be a feasible and practical option in alleviating sleep disturbance in this population. Further studies will be needed to ascertain FIT-IN’s effectiveness.

## 1. Introduction

Firefighters work in an environment that requires a rotating shift system, which leads to irregular sleep patterns and vulnerabilities to sleep disturbance and mental health issues, such as depression and suicidality [1,2,3]. In Korea, approximately 85.6% of firefighters are shift workers [4]. Shift workers may suffer from circadian misalignment [5] due to work and sleep schedules that conflict with the natural sleep-wake cycle [6]. In general, the rate of sleep problems in shift workers is higher than that of the general population [7]. A study by Ohayon [7] found that 20.1% of shift workers reported difficulty initiating sleep more than twice a week, which compared to 12% for dayworkers. In addition, only 28.5% of shift workers reported getting 4 hours or more of regular sleep, compared to 63.8% for dayworkers. Previous studies have reported that night and shift workers have lower sleep quality compared to day workers and report longer sleep onset latency due to pre-sleep high arousal [8]. Additionally, shift work increases the risk of insomnia, excessive sleepiness, fatigue, and shift work disorder [9]. According to a previous study with American firefighters, 37.2% of firefighters reported having at least one potential sleep disorder [10]. In a previous study with Korean firefighters, sleep efficiency and total sleep time were worse for shift workers compared to day workers [11]. Another study has suggested that insomnia symptoms in Korean firefighters may be associated with the frequency of fire suppression and emergency rescue [12]. 

Firefighters are repeatedly exposed to traumatic events, such as witnessing accidents, and frequently have close brushes with death, which increases the risk for PTSD (posttraumatic stress disorder) [13,14]. In practice, there is a 6.5%~37% prevalence of PTSD among firefighters [14,15,16,17,18,19,20]. One common symptom of PTSD is nightmares [21], with one preliminary study reporting a prevalence of 19.2% of firefighters experiencing nightmares [22]. In addition to shift work, nightmares are also the main cause of sleep disturbance in firefighters.

Brief behavior therapy for insomnia (BBTi) is a short-term sleep intervention that facilitates high patient compliance because of shorter delivery time compared to traditional cognitive-behavioral therapy for insomnia (CBTi) [23,24]. BBTi was developed to address common barriers, such as the duration of treatment sessions and the number of specialty-trained clinicians available that is encountered in the provision of evidence-based CBTi [25]. BBTi has the advantage of being effective with short-term sessions, and easier deliverability by a nurse or paraprofessional without a large amount of specialty training [26]. In addition, BBTi was developed to meet the needs for primary care settings, focusing on brief content and behavioral strategies, and comprised of stimulus control (an intervention that helps strengthen the association between the bed and sleep by instructing participants to use the bed for sleeping only), sleep restriction (an intervention that limits the time participants spent in bed awake), and some sleep-wake rules, such as limiting naps [24]. 

BBTi has been shown to be effective in improving insomnia in the elderly and treatment-resistant or comorbid patients [27,28,29]. Individuals who participate in BBTi are asked to follow simple and brief instructions, such as limiting time in bed for sleeping purposes only, or avoiding naps unless there are safety concerns. A summary of the content of BBTI can be found here [25]. BBTi has been validated by several previous studies, one of which found that individuals who received BBTi showed significant improvements for insomnia severity and sleep efficiency compared to the control group, who kept sleep diaries for self-monitoring compared to an active intervention [27]. The time efficiency, economic and remedial values of BBTi in various groups may be appropriate for firefighters who need long-term effectiveness of treatment results but may be unable to participate in long-term sessions due to work demands such as shift work and frequent dispatch. 

PTSD and nightmares require specific treatment because they lower sleep quality [30]. Imagery rehearsal therapy (IRT) is an intervention that uses imagery techniques to treat nightmares. IRT is an evidence-based therapy found to be effective in reducing nightmare frequency and intensity by rescripting nightmare content [31]. IRT has two therapeutic premises; one is that nightmares are learned sleep disorder, and the other one is that it is a symptom of a damaged imagery system. A previous study using IRT to reduce the frequency of chronic nightmares in patients who have PTSD found that the rate of nightmares decreased, with concurrent benefits for improving the quality of sleep improved and a decrease in the severity of PTSD symptoms [32]. Meta-analysis has validated that both IRT-only and combined with CBTi therapy sessions are effective in reducing the frequency of nightmares and improving sleep [33,34,35]. 

Currently, BBTi has not been used and tailored to firefighters. Therefore, the development of a sleep intervention that accounts for occupational characteristics such as frequent dispatch and shift work is needed. This study focused on developing a sleep intervention targeting improvement for sleep disturbance commonly found in firefighters (insomnia and nightmares). The ‘FIT-IN (Firefighter’s therapy for insomnia and nightmares)’ program consisted of three sessions (two face-to-face group sessions and one individual phone session), including sleep restriction and stimulus control, which are vital components of BBTi, with a nightmare rescripting component added from IRT. This pilot study was the first sleep intervention for firefighters via short-term intervention. We expected to see improvements in insomnia and nightmare severity, in addition to other psychological factors that have been shown to improve with sleep treatment, in firefighters through the ‘FIT-IN’ program.

## 2. Materials and Methods

### 2.1. Participants

This study was conducted as part of the sleep panel study (SLEPS) to improve sleep disturbance in Korean firefighters [36,37]. This study was conducted as part of a larger 7-year sleep panel study (SLEPS) to improve sleep disturbance in Korean firefighters [36,37]. The purpose of SLEPS was to investigate sleep disturbance among Korean firefighters and develop and investigate the efficacy of a sleep intervention program, with the final goal of nationwide dissemination. The current study was conducted in the 3rd year of SLEPS, with the goal of developing and investigating the efficacy of the FIT-IN intervention. For this study, participants were recruited from Korean fire stations from April to December 2019. Flyers and announcements about the sleep intervention were provided to fire stations, and 45 firefighters volunteered to participate in the study. Of the 45 firefighters, six dropped out of the intervention due to sudden dispatch during the session, absence for personal reasons, and tardiness. As a result, 39 firefighters from six fire stations (5 in Daejeon metropolitan city, 1 in Hwaseong, Gyeonggi) participated in the study.

### 2.2. Intervention

FIT-IN (Firefighter’s therapy for insomnia and nightmares) was developed based on BBTi, an evidence-based treatment effective in improving sleep disturbance [25,26,38]. IRT, an evidence-based treatment for nightmare disorders [31] was also added to address nightmares due to the high prevalence of PTSD in this population. A structured intervention manual for the therapist was developed. The intervention manual included detailed instructions required for each session as described in Table 1. 

This intervention consisted of a total of three 90-minute weekly sessions: two 90-minute face-to-face group sessions and one 20-minute telephone session. “Sleep education” was the main theme of the first session. Sleep education contained the following components: (a) 4 rules of BBTi (reduce time in bed, wake up at the same time of day every day no matter how poorly you slept the night before, do not go to bed unless you are sleepy, and do not stay in bed unless you are asleep [26]); (b) a two-process model of sleep; (c) sleep restriction and prescription of a sleep window, and; (d) behavioral strategies to cope with sleep problems in shift workers. Behavioral strategies to help cope with shift work included guidelines such as reducing unnecessary time spent in bed when awake on off-days, addressing the sleep environment when sleeping during the day after the night shift and keeping the room dark, wearing earplugs to shut out the noise, and wearing UV blocking sunglasses or goggles on the way home after the night shift. After sleep education, the therapist helped participants set individual sleep goals and taught them how to keep a sleep diary. While the content was delivered in a group format, individualized sleep schedules were provided to study participants to guide their sleep for the following week.

The theme of the second session was “nightmares”. At the beginning of the session, specific difficulties experienced by the participants in implementing the sleep goals or writing sleep diaries during the first week were covered. Additionally, diaphragmic breathing and relaxation therapy was introduced to the participants. Finally, the therapist educated participants about nightmares and conducted nightmare rescripting using an IRT protocol. In IRT, the use of vivid imagery helps the participants change their nightmares into new dreams.

The third and last session was conducted via telephone call as an individual session. The third session’s theme was “maintaining change”. A clinical psychologist and a participant spoke for about 20 minutes. The therapist addressed any difficulties the participant had in practicing the assignments during the second week. During this session, the therapist reinforced the efforts the participant made to change their sleep habits. The therapist and participant reviewed and discussed the extent to which the participant had achieved their goals. The therapist and participant explored ways to maintain sleep habits and utilized relapse prevention techniques.

### 2.3. Procedures

Prior to conducting the study, the research protocol was approved by the Institutional Review Board of Yonsei University Wonju Severance Christian Hospital (CR318031). Participants were informed of the nature and purpose of the study, and everyone signed informed consent forms. The first two sessions were conducted as a group (a minimum of four to a maximum of eight participants) by group facilitators. Group facilitators included the main therapist, who was a licensed clinical psychologist, and two master’s level student therapists who were oriented and trained in behavioral sleep medicine. The clinical psychologists led most of the main intervention, while the master's students helped participants in the first session to calculate indices from their sleep diary, such as sleep efficiency. In addition, the master's students created individualized sleep graphs based on the sleep diaries collected for two weeks to assist the clinical psychologists in providing the intervention.

In the first session, prior to intervention, the participants completed self-report questionnaires about demographic information and clinical indices (sleep and psychological factors). At the end of the second session, the participants completed the same questionnaires (excluding the questionnaire about demographic information). The telephone session was conducted as an individual session by a clinical psychologist and a participant. 

Six of the 45 registered participants dropped out during the intervention, with a total of 39 participants completing the intervention. The reasons for drop-out were as follows: tardiness (*n* = 2), absent after first session (*n* = 2), absent due to emergency dispatch (*n* = 1), and absent for personal reasons (*n* = 1).

### 2.4. Measures

#### 2.4.1. Sleep-Related Factors

##### Sleep Diary

Participants were informed of the nature and purpose of the study, and everyone signed informed consent forms. Participants completed a sleep diary for two weeks to record the following sleep parameters while participating in the intervention: sleep onset latency (SOL), number of awakenings (NWAK), wake after sleep onset (WASO), time in bed (TIB), total sleep time (TST), and sleep efficiency (SE). SE was calculated by dividing the TIB by the TST.

##### Insomnia Severity Index (ISI)

The ISI was a seven-item self-report measure of insomnia symptoms and related distress over the past month [39]. The measure consisted of 7 items targeting sleep disturbance severity, sleep-related satisfaction, degree of daytime functional impairment, impairment perception, and distress and concern related to the sleeping problem. Each item was rated on a 5-point Likert scale (0–4) and summed up to provide a total score ranging from 0 to 28. Higher scores reflected more severe insomnia symptoms. The cut-off for clinically significant sleep disturbance was 15 [40]. The internal consistency of this questionnaire was 0.80 in this sample. This questionnaire was validated in Korean [40]. 

##### Disturbing Dream and Nightmare Severity Index (DDNSI)

Disturbing Dream and Nightmare Severity Index (DDNSI) [41] was a 5 item self-report scale that was a revised version of the Nightmare Frequency Questionnaire (NFQ) [42]. The questionnaire assessed the frequency, intensity, and severity of nightmares. The scale was scored on a 0 to 37 scale, and higher scores reflected more severe nightmare difficulties. A total score greater than 10 was considered reflective of clinical levels of nightmares disorder [43]. This questionnaire was validated for use in Korean [44].

##### Korean version of the Epworth Sleepiness Scale (ESS)

The Epworth Sleepiness Scale (ESS) [45] was an 8-item self-report questionnaire that measured the tendency of sleep and sleepiness during the day in everyday situations. Participants were asked to respond to each question on a 4-point Likert scale, with each question ranging from 0–3. A total score was calculated by summing up all items. Higher scores corresponded to higher extreme daytime sleepiness. The internal consistency of this questionnaire was 0.74 in this sample. This questionnaire was validated in Korean [46]. 

#### 2.4.2. Psychological Factors

##### Patient Health Questionnaire-9 (PHQ-9)

The PHQ-9 [47] consisted of nine questions designed to correspond to the nine diagnostic criteria for major depressive disorder covered in the Diagnostic and Statistical Manual of Mental Disorders (DSM–IV). The nine items were rated from 0 to 3 according to the increased frequency of experiencing difficulties in each area covered. Scores were totaled and ranged from 0 to 27. The score could then be interpreted as indicating either no depression (1 to 4), mild (5 to 9), moderate (10 to 14), moderately severe (15 to 19), or severe depression (20 to 27). The optimal cut-off as a depression screening tool was 10 [48]. The internal consistency of this questionnaire was 0.88 in this sample. This questionnaire was validated in Korean [49].

##### PTSD Checklist-5 (PCL-5)

The PCL-5 was used to measure PTSD and was a 20-item self-report measure that evaluated the degree to which an individual was bothered in the past month by his or her most current distressing event as defined by DSM–5. Items were rated from 0 (not at all) to 4 (extremely) and were added up for a total severity score [50]. Severity scores were calculated by summing up items in each of the four DSM–5 PTSD symptom clusters: intrusions (items 1–5), avoidance (items 6–7), negative alterations in cognitions and mood (items 8–14), and alterations in arousal and reactivity (items 15–20). PTSD was considered as endorsing a severity of at least a 2 (moderate) for a sufficient number of symptoms in each cluster to meet DSM–5 criteria. The internal consistency of this questionnaire was 0.96 in this sample. This questionnaire was validated in Korean [51].

##### Depressive Symptom Inventory-Suicidality Subscale (DSI-SS)

The DSI-SS [52] consisted of 4 items and was a subscale of the Hopelessness Depression Symptom Questionnaire. Each question was measured on a 4-Point Likert scale 0–3, with total scores ranging from 0 to 12. The scale assessed the frequency of suicidal ideation, specific plans for suicide, controllability of suicidal thoughts, and impulses in the previous 2 weeks. Higher total scores indicated greater severity of suicidal ideation [53]. In this study, the cut-off of this questionnaire was based on 4 scores suggested by the validated Korean version of DSI-SS [54]. The internal consistency of this questionnaire was 0.93 in this sample.

### 2.5. Data Analysis

SPSS 21.0 (SPSS Inc., Chicago, IL, USA) was used for data analysis. We conducted frequency analysis and descriptive statistics for demographic information. For the main analysis, the Wilcoxon signed-rank test or paired t-tests were used to compare the results of the measures for the participants before and after the FIT-IN program. For the DDNSI data, unlike the other data, a normality test was performed first, because the sample size was 10 ≤ *n* < 30 (pre: 20, post: 17). The assumptions of normality were not met, so the Wilcoxon signed-rank test was performed. The test interpreted that the difference between before and after treatment was statistically significant if the null hypothesis could be rejected, with a *p*-value < 0.05. 

For the sleep diary, the first week’s data were analyzed as preliminary data, and the second week’s data was post data. To use the data that had established the representativeness of each week, individuals who completed less than three days of the sleep diary were excluded [55]. For clinical indicators, responses at the beginning of session 1 were used as preliminary data, and data at the end of session 2 were used as post data. Missing data were not imputed for statistical analysis in this study.

## 3. Results

### 3.1. Demographics and Sample Characteristics

Demographic information of the participants was collected at the time of the baseline. The mean age of the participants was 43.33 (±9.32) years. Of the total 39 participants, 84.6% (*n* = 33) were male. Participants had been employed for an average of 9.98 (±7.62) years. Among the participants, 61.5% (*n* = 24) reported currently being a shift worker for an average of 16.54 (±10.14) years. All shift workers had a rotating three-shift system with a 21-day-cycle, consisting of one week of day work and two weeks of night shifts. All non-shift participants were day workers (*n* = 15; 38.5%). General characteristics are summarized in Table 2.

### 3.2. Intervention Effects on Sleep-Related Factors

The Wilcoxon signed-rank test and paired t-test analysis showed that the intervention had significant effects on improving sleep by reducing SOL (pre to post: 33.51 ± 34.79 vs. 21.79 ± 21.80 minutes; *t* = 2.063, *p* = 0.048), NWAK (pre to post: 0.88 ± 1.11 vs. 0.58 ± 1.00; *t* = 2.564, *p* = 0.015). Sleep efficiency was improved significantly (pre to post: 80.87 ± 12.97 vs. 87.33 ± 10.05; *t* = −3.036, *p* = 0.005) along with TIB (pre to post: 6:50 ± 0:48 vs. 6:30 ± 0:38 hours; *t* = 2.158, *p* = 0.39). Additionally, insomnia severity (ISI; pre to post: 15.33 ± 4.78 vs. 10.87 ± 4.86; *t* = 7.209, *p* < 0.001), nightmare severity (DDNSI; pre to post: 5.08 ± 6.74 vs. 1.75 ± 4.51; *z* = −2.926, *p* = 0.003), and excessive daytime sleepiness (ESS; pre to post: 8.51 ± 4.05 vs. 7.05 ± 3.28; *t* = 2.517, *p* = 0.016) were all significantly decreased. 

The proportion of participants with an ISI score above the clinical cut-off (score of 15) was 53.8% (*n* = 21) at baseline. Following intervention, the rate for remission was 76.19% (*n* = 16). Additionally, 15.4% of participants scored above the clinical cut-off on the ISI after the intervention.

Details are described in Table 3.

### 3.3. Intervention Effects on Psychological Factors

Results indicated that depression (PHQ-9; pre to post: 6.79 ± 5.09 vs. 4.67 ± 4.65; t = 3.71, *p* < 0.001) and PTSD symptoms (PCL-5; pre to post: 16.33 ± 15.58 vs. 10.26 ± 11.85; t = 2.769, *p* = 0.009) were significantly reduced after the FIT-IN program. Details are presented in Table 3.

## 4. Discussion

The aim of the current study was to develop a sleep intervention tailored to firefighters based on an evidence-based treatment (BBTi). Preliminary results indicated that FIT-IN, a brief intervention developed for the purposes of our study, significantly improved sleep indices such as decreasing sleep onset latency and improving sleep efficiency. In addition, significant improvements were found for insomnia severity, depression, post-traumatic stress disorder symptoms, nightmare severity, and daytime sleepiness. This is the first preliminary study, to the best of our knowledge, to investigate the effects of brief-behavioral therapy for insomnia combined with imagery rehearsal therapy in firefighters. 

Previous studies have investigated a sleep health program (SHP) for firefighters. Sullivan and colleagues conducted a study to determine if the sleep health program was effective against dangerous injuries and disabilities for firefighters [56]. This intervention consisted of three parts: (1) mandatory educational sessions, including training on fatigue-related health hazards and the physiological importance of sleep for about 30 min; (2) voluntary sleep disorders screening that identified sleep disorders such as insomnia; (3) diagnosis of sleep disorders and treatment for firefighters who reported positive for one or more sleep disorders, including obstructive sleep apnea, insomnia, restless legs syndrome, and shift work disorder. The group receiving SHP through an intervention reported fewer days of disability compared to the control group. Programs that provided sleep health education and sleep disorder testing opportunities could reduce injuries and work losses due to the disability of firefighters. In another study, a sleep health program that included a 30-minute sleep education training session with the content on healthy sleep and fatigue countermeasures was conducted for firefighters [57]. The SHP was delivered utilizing three methods: expert-led, train-the-trainer, and online. As a result of the study, firefighters reported the importance of the SHP itself, despite the method of delivery. Similar to firefighters, when fatigue management training was performed for police officers working long work hours and shift work, a significant improvement was found in insomnia and sleep satisfaction after the training [58]. Fatigue management training was conducted to improve the sleep health of police officers, and participants learned basic sleep needs, and methods to identify symptoms of sleep disorders, and methods to combat fatigue and improve sleep hygiene. Previous studies have reported that providing firefighters with opportunities for sleep education and sleep disorder diagnosis and treatment has a positive effect on firefighters and that the sleep health program itself is important. Although FIT-IN did not diagnose or provide intervention for sleep disorders, this study adds to the literature that sleep interventions for firefighters are indeed effective and help improve other domains that are associated with the occupation.

The intervention in this study was developed in consideration of the work environment of firefighters. First, firefighters frequently are on call for unpredictable events such as emergency dispatch and fire suppression. Due to these occupational characteristics, firefighters may be limited to participating in an intervention consistently for an extended period of time and therefore require shorter sessions in an intervention. Thus, this study consisted of a total of three sessions (two in-person, one telephone call) by modifying the BBTi of the existing four sessions. Second, the current study added sleep guidance for the firefighters who participated in shift work. In a previous study that examined factors related to sleep disorders in firefighters, 78.2% of the 657 firefighters were found to be working in shifts [59]. This may lead to desynchronization in circadian rhythm and sleep deprivation [60,61]. The current study added sleep guidelines for shift workers based on previous research [62,63,64], such as wearing UV-blocking sunglasses after working the night shift when returning home, and timing naps and light exposure in order to tailor the intervention to be more helpful to firefighters. Third, firefighters may be hypervigilant to emergency cues when they are on duty, rendering it difficult to de-arouse when they return home from work [65]. Considering that high levels of hyperarousal are a core mechanism of insomnia [66], relaxation techniques to help them lower arousal prior to sleeping were emphasized in the intervention. Finally, firefighters are frequently exposed to situations that can cause post-traumatic stress [67], and a previous study showed that 19.2% of firefighters suffered from nightmares [57]. Thus, a component of IRT, an evidence-based treatment for nightmares, was added to FIT-IN.

Preliminary, results from the current study showed promising results for both sleep and clinical indices. SOL decreased from 33 min to 21 min, and post-intervention average SOL was reported to be within normal limits [68]. This was consistent with previous studies that showed significant improvement in sleep indices following BBTi in various target groups such as the elderly and adult insomnia patients [23,24,28]. There were also improvements for insomnia severity and daytime sleepiness scores, suggesting that brief intervention is helpful in alleviating sleep disturbance in this population. Participants also showed improvements in depression and PTSD symptoms. Although FIT-IN was a sleep-focused intervention, it was also effective in alleviating psychological factors, which was consistent with a previous study implementing BBTi in veterans. In this study, BBTI also had a significant effect on decreasing both insomnia and depression symptoms [69]. Another study improved quality of sleep and reduced PTSD symptoms through a single session of treatment combining IRT through a short-term behavioral intervention in adult victims of violent crimes who were currently diagnosed with PTSD [31]. Thus, there may be advantages to mental health when receiving a sleep intervention in firefighters. 

It is important to note that this is the first sleep intervention developed in Korea for firefighters tailored to their occupational characteristics and working environment. FIT-IN was developed to be a short-term intervention that combines BBTi and IRT components to help improve aspects of sleep that are often found in this population. Previously, many studies have been conducted on the effectiveness of an intervention that combines CBTi and IRT [33,34,35], but there were no studies on programs that combine BBTi and IRT. Results were promising in that participants' sleep indices, nightmare severity, depression, and PTSD symptoms all significantly improved following the intervention. Despite the brief intervention consisting of three sessions, sleep and mental health also improved after participating in FIT-IN. Based on the results of this current study, it showed that FIT-IN is a sleep intervention that is feasible for firefighters. 

### Limitations

The limitations of the current study and suggestions for future research are as follows. First, as a pilot study, this study focused on the development of the intervention, and thus implemented a single-group pre-post design. The absence of a control group makes it difficult to strongly determine the effectiveness of the FIT-IN intervention. Future studies recruiting larger sample sizes and implementing a randomized controlled trial are needed to ascertain intervention effectiveness. 

Second, this was a pilot study with a small sample size, thus the results have limits to generalizability. Males represented 84.6% of the sample, therefore, the sample may have limited gender variation. However, considering that about 91% of Korean firefighters were male in 2019 [70], it can be seen that this study data well represents the gender of firefighters in Korea. In addition, the results should be interpreted with caution when generalizing the results to other areas of Korea and in other countries. 

Third, this study did not use objective measurement tools to measure sleep, such as actigraphy or polysomnograms. There was no assessment for other sleep disorders such as obstructive sleep apnea, which may have been pertinent to this population of middle-aged men. Further studies will be necessary to explore various reasons contributing to sleep disturbance in firefighters using both subjective and objective measurements.

Fourth, FIT-IN was performed with three sessions, but the post-test was performed at the end of the second session. Therefore, the effect of the third session could not be measured. In addition, since sleep diaries were collected from the start of the intervention so it was difficult to interpret the sleep status of firefighters before the intervention. This was especially difficult in light of the fact that there were no specific selection criteria for participating in the study aside from being a firefighter. In a future study, it is necessary to collect sleep diaries for a week before the intervention begins, to check the sleep status of firefighters, use more stringent inclusion criteria, and to compare follow-up data by conducting inspections by time point.

Finally, intervention fidelity was not measured in this study. While the therapists used a manualized treatment to guide the sessions, adherence to the treatment manual was not assessed. Future studies should include fidelity measures and investigate whether treatment fidelity corresponds to better intervention effects. Similarly, intervention adherence data was not collected from the participants, thus it is difficult to conclude that close adherence to the intervention leads to better outcomes. 

## 5. Conclusions

Firefighting is a primary service that is directly connected to the lives and safety of the community, and the continuous and efficient provision of firefighting services requires continuous intervention for firefighters. More than 70% of firefighters had at least one sleep problem [71], and sleep problems can lead to reduced work ability and shorter life expectancy in the long term, so sleep interventions for firefighters are important and necessary. The BBTi-based sleep intervention developed in this study was the first intervention developed in Korea that improved sleep disturbance in a relatively short period of time for firefighters through a structured and manualized treatment protocol. 

## Figures and Tables

**Table 1 ijerph-17-08738-t001:** Session-by-session protocol of Firefighter’s Therapy for Insomnia and Nightmares (FIT-IN).

Session	Theme	Goals and Activities	Homework
1st Group Session	Sleep Education	Sleep education 4 rules of BBTi Two-process model of sleep, 3-P model of insomnia Sleep restriction Behavioral strategies to cope with sleep problems in shift workers Set individual sleep goals Keeping a sleep diary	Sleep Diary
2nd Group Session	Nightmare	Checking sleep diary and adjusting sleep window (sleep restriction) Practicing breathing and relaxation therapy Psychoeducation about nightmares and nightmare rescripting	Sleep Diary
3rd Private Telephone Session	Maintaining Change	Review sleep restriction, sleep diaries and practice relaxation Review treatment goals Discuss relapse prevention strategies	

**Table 2 ijerph-17-08738-t002:** Demographics and sample characteristics (*n* = 39).

Variables		*N* (%) or M (±SD)
Sex	Male	33 (84.6)
	Female	6 (15.4)
Working type	Shiftwork	24 (61.5)
	Daywork	15 (38.5)
* Rank	3rd~5th Firefighter	5 (12.8)
	1st~2nd Firefighter	5 (12.8)
	Lieutenant	11 (28.2)
	Captain	9 (23.1)
	Battalion Chief	7 (17.9)
	Deputy Chief	2 (5.1)
Years of Service (year)		9.98 ± 7.62
Duration of Shift Work (year)		16.54 ± 10.14

* Ranks: 3rd~5th Firefighter, 1st~2nd Firefighter: basic grade firefighter; Lieutenant: Officer of a single team (company) and/or a fire station; Captain: Head of safety center team; Battalion Chief: Fire department team leader or director of the safety center; Deputy Chief: Head of municipal and provincial headquarters, chief of fire department.

**Table 3 ijerph-17-08738-t003:** The difference of indices between before and after FIT-IN (*n* = 39).

Variables	Index	Before FIT-IN (M ± SD)	After FIT-IN (M ± SD)	*t*/*z*^†^	*p*	Effect Size
						*d*
Sleep-related factors (*n* = 32)	SOL (min)	33.51 ± 34.79	21.79 ± 21.80	2.063	0.048 *	0.36
	NWAK (n)	0.88 ± 1.11	0.58 ± 1.00	2.564	0.015 *	0.45
	WASO (min)	27.47 ± 29.21	17.09 ± 20.99	1.676	0.104	0.30
	TIB (hh:mm)	6:50 ± 0:48	6:30 ± 0.38	2.158	0.039 *	0.38
	TST (hh:mm)	5:32 ± 0:53	5:39 ± 0:47	−0.824	0.416	0.15
	SE (%)	80.87 ± 12.97	87.33 ± 10.05	−3.036	0.005 **	0.54
	ISI (*n* = 39)	15.33 ± 4.78	10.87 ± 4.86	7.209	0.000 ***	1.15
	DDNSI ^†^ (*n* = 16)	5.08 ± 6.74	1.75 ± 4.51	−2.926 ^†^	0.003 **	
	ESS (*n* = 39)	8.51 ± 4.05	7.05 ± 3.28	2.517	0.016 *	0.40
Psychological factors (*n* = 39)	PHQ−9	6.79 ± 5.09	4.67 ± 4.65	3.715	0.001 **	0.59
	PCL-5	16.33 ± 15.58	10.26 ± 11.85	2.769	0.009 **	0.44
	DSI-SS	0.51 ± 1.35	0.21 ± 0.923	1.966	0.057	0.31

* *p* < *0*.05, ** *p* < 0.01, *** *p* < 0.001. Statistical analysis was done by the Wilcoxon signed-rank test (†) and the paired t-test. Effect sizes for all outcomes (except DDNSI) were calculated using Cohen’s *d.* Abbreviations: SOL = Sleep onset latency; NWAK = Number of awakenings; WASO = Wake after sleep onset; TIB = Time in bed; TST = Total sleep time; SE = Sleep efficiency, ISI = Insomnia severity index; DDNSI = Disturbing dream and nightmare severity index; ESS = Epworth sleepiness Scale; PHQ-9 = Patient health questionnaire-9; PCL-5 = PTSD checklist for DSM-5; DSI-SS = Depressive symptom inventory-suicidality subscale.

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
