# Peer review of "The Development of a Sleep Intervention for Firefighters: The FIT-IN (Firefighter’s Therapy for Insomnia and Nightmares) Study"

_ijerph, 2020, doi:10.3390/ijerph17238738_

Round 1
Reviewer 1 Report
The present study examined the effects of brief behavioral therapy for insomnia combined with imagery rehearsal therapy in firefighters. As a preliminary study, there were several methodological limitations, e.g. a lack of control group and the timing of the assessments, which were well acknowledged by the authors. Nonetheless, the findings of this study have important clinical/practical implications given that firefighters represent a unique group that is vulnerable to sleep and mental health problems. This study provided important pilot data to lay the foundation for future rigorous trials.
Introduction:
In the research background, it would be helpful to mention other mental health problems in this unique group, apart from PTSD, so as to justify several psychological factors (e.g. depression) assessed in this study.
This study included several psychological measures (e.g. mood symptoms, PTSD symptoms, suicidality) but these were not mentioned in the study objectives/hypotheses.
Methods:
Line 117: can the authors provide the information on the four rules of BBTi?
Line 115 and line 128: It was stated that the intervention consisted of a total of three 90-minute weekly sessions, but later in line 128 it was stated that a clinical psychologist and a participant spoke for about 20 minutes in the individual session. Can the authors clarify whether the individual session lasted for 90-min or 20-min?
Line 153: This sentence is the same as 136 & 137: Participants were informed of the nature and purpose of the study, and everyone signed 153 informed consent forms.
Did the intervention introduce any behavioral strategies to address how to cope with sleep problems due to shift work schedules? It would be helpful to mention these in the methodology section.
Table 2: can the authors provide some explanations about the ranking listed in this table?
Table 3: It would be helpful to provide the effect sizes.
In terms of insomnia, what was the rate for remission and response to intervention?
Discussion:
Given the limited sample in this study, it would be helpful to interpret the effect size and also response/remission rates of insomnia.
Line 310: please indicate in what population the study (reference 27) was conducted.
Minor grammatic issues:
Line 70: ‘...some sleep-wake rule(s).’
Line 119: There was a grammatical mistake: ‘the therapist helped participants set individual sleep goals and [taught them] how to write a sleep diary.’
Line 121: ‘difficulties experienced by the participant[s] in implementing…’
Reviewer 2 Report
This is a very interesting paper. This is a first kind of study, given lack of intervention studies targeting sleep issues among firefighters, particularly in Korea. Preliminary results are very promising.
Abstract
-Include the study design
-Indicate sleep variables were from sleep diaries.
-Address “other clinical indices”
Introduction
-Page 2, line 51, “In general, the rate of sleep problems is higher than that of the general population in shift workers”. Add the data.
-Authors interchangeably used the term, “BBTI” and “BBTi”.
-Page 2, line 65, change “cognitive behavior therapy” to “cognitive behavioral therapy” and “CBT” to “CBTi”
-Page 2, line 70, briefly explain what “stimulus control, sleep restriction” are. Also, what “some sleep-wake rule” means. Even a very brief explanation would help readers to understand these key components of CBTi. Same for sleep efficiency- add its definition using the parentheses.
-Page 2, line 77, what type of control group was compared?
-Page 2, line 95, revise “three sessions in a group therapy format” to “three sessions (two face-to-face group sessions and one individual phone session)”
-Was any study looking at sleep among Korean firefighters? If so, add some key findings.
Methods
-page 3, authors said that this study was a part of the SLEPS. How is the FIT-IN different from the SLEPS? Further explanation of the parent study and its detailed goal need to be added.
-Study eligibility criteria is missing (both inclusion and exclusion). Were participants from the SLEPS excluded from the FIT-IN? I think both poor sleep and having some symptoms of PTSD should be a part of inclusion criteria, however, if not, address this as a limitation.
-Sleep restriction is not clearly addressed. Was sleep restriction actually delivered? If so, add it under Intervention section. I believe that this component should have been delivered in session 2. Were individualized sleep schedules provided to study participants in the setting of group sessions?
-Page 3, Table 1, list “4 rules of BBTi”.
-Group facilitators included a clinical psychologist and two Master’s students. Add role of each facilitator for the intervention.
-This intervention consists of two face-to-face group session and one “individual” phone. Please add this information clearly.
-Was any intervention fidelity measured? If not, include this as limitation.
Measures
-Page 4, line 153, “Participants were informed of the nature and purpose of the study, and everyone signed informed consent forms” is redundant. Remove it.
-I assume that all standardized instruments were administered in Korean. Add the citations for each Korean version in addition to the original one.
Results and Discussion sections were overall well written.
-Were intervention adherence data collected? Such as how many N completed what the researchers delivered? For example, N who followed sleep schedules etc.
-In limitations, add that the findings of this study cannot be generalized in other areas of Korea and in other countries.
